# Clinical Manifestations and Causes of Anaphylaxis. Analysis of 382 Cases from the Anaphylaxis Registry in West Pomerania Province in Poland

**DOI:** 10.3390/ijerph17082787

**Published:** 2020-04-17

**Authors:** Iwona Poziomkowska-Gęsicka, Michał Kurek

**Affiliations:** Clinical Allergology Department, Pomeranian Medical University (PMU) in Szczecin, Powstańców Wlkp 72, 70-111 Szczecin, Poland; allergy@pum.edu.pl

**Keywords:** anaphylaxis, cause, clinical symptoms, epidemiology

## Abstract

Anaphylaxis is most commonly defined as an acute, severe, potentially life-threatening systemic hypersensitivity reaction. Current expert consensus has defined anaphylaxis as a serious reaction that is rapid in onset and can be fatal, and is a severe, potentially life-threatening systemic hypersensitivity reaction that is still rarely diagnosed. For safety reasons, patients should visit an allergologist to identify potential causes of this reaction. There are no data from other health care centres in Poland presenting characteristics of anaphylactic reactions. Clinical manifestations of anaphylaxis should be analysed, because some patients (10–30%) with anaphylaxis can present without cutaneous findings. This lack of skin/mucosa involvement can lead to misdiagnosis or delayed diagnosis of anaphylaxis. Objectives—to gather epidemiological data on anaphylactic reactions, to identify clinical manifestations of anaphylaxis (organ systems involved), to present diagnostic methods useful for the identification of anaphylaxis triggers, and most importantly, to find causes of anaphylaxis. In this retrospective analysis, we used a questionnaire-based survey regarding patients visiting the Clinical Allergology Department, Pomeranian Medical University (PMU) in Szczecin, between 2006 and 2015. The registry comprised patients with grade II (Ring and Messmer classification) or higher anaphylaxis. Patients with grade I anaphylaxis (e.g., urticaria) were not included in the registry. The incidence of anaphylaxis was higher in women. Clinical manifestations included cutaneous and cardiovascular symptoms, but more than 20% of patients did not present with cutaneous symptoms, which may create difficulties for fast and correct diagnosis. Causes of anaphylaxis were identified and confirmed by means of detailed medical interview, skin tests (STs), and measurement of specific immunoglobulin E (sIgE) and tryptase levels. In the analysed group, the most common cause of anaphylaxis (allergic and nonallergic) was *Hymenoptera* stinging (wasp), drugs (nonsteroidal anti-inflammatory drugs, NSAIDs) and foods (peanuts, tree nuts, celery). The incidence of anaphylaxis is low, but because of its nature and potentially life-threatening consequences it requires a detailed approach. Comprehensive management of patients who have had anaphylaxis can be complex, so partnerships between allergy specialists, emergency medicine and primary care providers are necessary. Monitoring its range is very important to monitor changes in allergy development.

## 1. Introduction

Anaphylaxis is most commonly defined as an acute, severe, potentially life-threatening systemic hypersensitivity reaction [1,2]. Current expert consensus has defined anaphylaxis as a serious reaction that is rapid in onset and can be fatal [3]. The “immediate” reaction described Gell and Coombs (as an allergic anaphylaxis) is when clinical signs appear within 2 h of exposure. But we must consider 2009, when the immediate hypersensitivity reaction (anaphylaxis to red meat) involved first symptoms that were not observed until 3−6 h after consumption. The signs and symptoms of acute anaphylaxis versus delayed anaphylaxis are similar [4]. Onset of anaphylaxis to stings or allergen injections is usually rapid—70% begin in less than 20 min, and 90% in less than 40 min [5]. Considering clinical symptoms, reaction involve at least two organ systems—cutaneous and cardiovascular, cutaneous and respiratory, or cutaneous and gastrointestinal [6]. Reduced blood pressure (compared to the baseline value for a specific age group) after exposure to a known allergen may also be a symptom of an anaphylactic reaction [7]. The classification by Johansson [8] indicates two mechanisms of hypersensitivity—immunologic (allergic, including IgE-mediated and not IgE-mediated reactions) and nonimmunologic. In the immunologic IgE-mediated mechanism, cross-linking of cell surface high-affinity IgE receptors (FcɛRI) initiated by the binding of IgE with antigens causes degranulation of effector cells. In the immunologic non-IgE-mediated mechanism, mast cells and basophils are activated by anaphylatoxins C3c and C4, after exposure to blood products or other agents activating the coagulation and complement cascade. In the nonimmunologic mechanism, involving a “nonallergic anaphylaxis”, also known by the older term “anaphylactoid reaction”, degranulation of effector cells in anaphylaxis can be caused by exercise, low temperature, and certain drugs [9,10,11].

Identification of the cause of anaphylaxis is important for the patient, while understanding the mechanism of reaction is also vital for clinicians. With detailed information available, it is possible to predict, for example, whether there are other agents that may trigger anaphylaxis (responsible for cross-reactivity, etc.).

Several systems are used to classify the severity of anaphylaxis, but the two presented below (Table 1) are the most popular.

The clinical symptoms of anaphylaxis are associated with the distribution of effector cells. Cutaneous symptoms include urticaria, angioedema, generalised erythema and itching. Respiratory symptoms include breathing disorders, dyspnoea, bronchospasm, laryngeal oedema, tongue swelling, and speech disorders. Patients may also present with cardiovascular symptoms, including tachycardia, bradycardia, drop in blood pressure, arrhythmia, acute coronary event, and/or gastrointestinal symptoms, including nausea, vomiting, diarrhoea, spastic abdominal pain, dysphagia, metallic taste in the mouth, and other symptoms, including acute rhinitis, conjunctivitis, uterine cramps, headache and dizziness [9]. Causes of anaphylaxis are various, and it is not always possible to confirm the suspected causal relationship. The most common causes of anaphylaxis, depending on the geographic region, include hypersensitivity/allergy to *Hymenoptera* venom, hypersensitivity/allergy to foods, and hypersensitivity/allergy to drugs [13]. In patients with food allergies, a correlation between the allergic reaction, the onset of symptoms and the degree of exposure to a given allergen has been observed. Foods popular in the diet of a given population create a greater risk of anaphylaxis than foods consumed occasionally (e.g., there is a high incidence of anaphylaxis after exposure to peanuts in the United States of America, marine fish in Scandinavia, seafood in Japan, protein from cow’s milk and hen eggs in Europe) [9]. The risk of anaphylaxis is increased because of faster introduction of new foods in a child’s diet, immaturity of the intestinal barrier in the youngest children, and increasingly varied diet. New sources of food proteins and the development of new technologies in food production change the immunogenic or/and allergenic potential of final product ingredients. Potential food allergens include cow’s milk, hen egg white, peanuts, wheat, soybeans, fish, celery, and seafood.

Anaphylactic reactions are also caused by popular drugs. The highest number of adverse reactions, including nonallergic and allergic anaphylaxis, are induced by nonsteroidal anti-inflammatory drugs (NSAIDs) and antibiotics. Depending on authors and various populations analysed, most reactions are either attributed to NSAIDs [14,15,16] causing nonallergic anaphylaxis, or beta-lactam antibiotics, being the most common cause of IgE-mediated allergic anaphylaxis [17]. Beta-lactams are also known to cause late-onset hypersensitivity reactions, e.g., maculopapular eruption [18], which should not be linked with anaphylaxis. Usually, less than 20% of all adverse drug reactions have an immunologic mechanism. A similar incidence was found after the verification of data on ‘allergic reactions’ to penicillin commonly reported by patients [19]. Considering nonsteroidal anti-inflammatory drugs (NSAIDs), potential anaphylactic reactions in the IgE-mediated mechanism were only documented for pyrazolones [20], and according to some authors, this class of NSAIDs most frequently causes anaphylaxis [21]. Other NSAIDs act in the nonallergic mechanism, primarily by inhibiting cyclooxygenase 1 [22,23], thereby causing cross-reactive hypersensitivity to substances of different chemical structure but having the same mechanism of action [23].

The objective of this work was to gather epidemiological data on anaphylactic reactions, to identify clinical manifestations of anaphylaxis (organ systems involved), to present diagnostic methods useful for the identification of anaphylaxis triggers, and most importantly, to find causes of anaphylaxis.

## 2. Material and Methods

### 2.1. Study Design and Data Collection

For the retrospective analysis, we used a questionnaire-based survey carried out by allergology specialists during the patient’s first visit in our centre. Of all 10,738 new patients examined at the Allergology Department in 2006–2015, with suspicion any allergic or nonallergic hypersensitivity, we found 490 patents with suspicious for moderate and severe anaphylaxis. After 1 year since the first visit, doctors analysed survey again, as well as additional results. Finally, we found that there were 382 cases of moderate and severe anaphylaxis (grades II-IV Ring and Messmer classification). Patients with grade I anaphylaxis were not entered in the registry.

The Basis Questionnaire-Simplified Version of the Network for Online Registration of Anaphylaxis Survey (NORA) from Berlin

(1)Did Your Patient, After Contact with Any Factor, Experience Any of the Following Symptoms?(2)Did your patient immediately experience any of the following symptoms without a reason? Difficulty breathing, wheezing, hypotension, cramping abdominal pain, diarrhoea, vomiting, loss of consciousness? If so, complete the data:
Patient’s Year of BirthDate of the Onset of AnaphylaxisPlace of ReactionGender(3)Mark the Organ Systems Involved:
Cutaneous Symptoms: Angioedema, Flush, Generalized Erythema, Generalized Itching, Generalized UrticariaRespiratory Symptoms: Apnoea, Dyspnoea, StridorGastrointestinal Symptoms: Abdominal Pain, Diarrhoea, Nausea, Vomiting, IncontinenceCardiovascular Symptoms: Loss of Consciousness, Drop in Blood Pressure, Collapse, Cardiac Arrest, Dizziness, Tachycardia, Disorientation
(4)Mark the Diagnostic Tests Used:
Medical interview, skin tests, sIgE, tryptase, provocation, other.
(5)Did the Reaction Occur for the First Time?(6)Is the Trigger Factor Known? Has the Trigger of Anaphylaxis been Identified?
Is this medicine? What is it exactly?Is this food? What is it exactly?Is this venom? What kind of venom?Other
(7)Have any Prevention Methods been Undertaken?(8)Please Write other Important Details about this Episode

### 2.2. Diagnostic Methods

The following diagnostic tests were used:(1)Skin Prick Tests (Allergopharma GmbH &CoKG/ Germany), Skin Prick Tests/Intradermal Tests for Medicine: penicilloyl polylysine (PPL) and minor penicillin determinant (MDM) DIATER/Spain; Amoxicillin, Ampicillin, and Cephalosporin (nonirritating concentration)-Accordance with European Network for Drug Allergy (ENDA) Recommendations.(2)Specific IgE (Omega Diagnostics GmbH/Germany and UniCap 100 /Phadia/Sweden)(3)In Selected/Specific Cases We Performed Basophil Activation Test (BAT)-Expression of CD203c (Beckman Coulter CD/United States).

### 2.3. Statistical Analysis

Obtained data were analysed using the Statistica 12 software package (StatSoft, Inc, Cracow, Poland license, Tulsa, USA). A basic statistics panel was used for data processing (descriptive statistics). Due to the qualitative features used in the analysis, nonparametric tests were used. For comparison between groups, the U Mann–Whitney test was used. The collected data were presented in the form of a multidivisional table, and for qualitative variables we used Pearson’s chi-square test. Statistical significance was adopted at a *p* value of *p* < 0.05.

## 3. Results

### 3.1. Descriptive Statistics

#### 3.1.1. The Incidence of Anaphylaxis in the Analysed Group

Of all 10,738 new patients examined at the Allergology Department in 2006–2015, there were 382 cases of moderate and severe anaphylaxis (grades II-IV by Ring and Messmer classification), which accounted for 3.56% of new patients. The incidence of anaphylaxis in the population of Western Pomerania province in the analysed 10-year period was in the range of 0.001–0.003%, with a mean annual incidence of 0.0021%.

The structure of research, with regard to the phases of the research and the division into groups, is shown in Figure 1.

#### 3.1.2. Gender in the Analysed Group

There were 208 women (54.4%) and 123 men (32.2%) with anaphylaxis, *p* < 0.000, and 51 children with anaphylaxis (13.4%).

#### 3.1.3. Age in the Analysed Group

The mean age in whole group was 40.5 years. The mean age at the onset of anaphylaxis was 45.3 years (range 19–80) for women (n = 208), and 44.3 years (range 19–79) for men (n = 123). In the group of children and adolescents (maximum age, 18 years) the mean age at onset was 11.3 years (9.9 for boys and 12.1 for girls).

### 3.2. Clinical Symptoms of Anaphylaxis

Presents in Figure 2.

#### 3.2.1. Organ Systems Involved 

(1)Cutaneous symptoms (angioedema, flush, generalized erythema, generalized itching, generalized urticaria) were found in 294 patients (76.96%);(2)Cardiovascular symptoms (loss of consciousness, drop in blood pressure, collapse, cardiac arrest, dizziness, tachycardia, disorientation) were found in 279 patients (73.04%);(3)Respiratory symptoms (apnoea, dyspnoea, stridor) were found in 258 patients (67.54%); and(4)Gastrointestinal symptoms (abdominal pain, diarrhoea, nausea, vomiting, incontinence) were found in 111 patients (29%).

The incidence of skin manifestations in children was statistically higher than in women (*p* = 0.0488) and children had more gastrointestinal symptoms than men (*p* = 0.0138) (Figure 3).

#### 3.2.2. Affected Organ Systems

The most common number of organ systems involved in whole group was two or three (Figure 4), compared to one or four organ systems (*p* = 0.000).

Organ systems involved (%) in anaphylaxis with a group division: women, men, and children.

Clinical manifestations involving one organ system was less common in children than women (*p* = 0.044) and men (*p* = 0.033). Four organ systems were involved more frequently in children than in women (*p* = 0.032). This relationship is shown in Figure 5.

Diagnostic methods used to identify the causes of anaphylaxis in the analysed group.

Medical interview—382 patients (100%);Skin prick tests (or/and intradermal tests)—282 patients (73.82%);Measurement of sIgE level—238 patients (62.3%);Measurement of tryptase level—148 patients (38.74%); andChallenge tests—23 patients (6.02%)

### 3.3. Triggers of Anaphylaxis:

#### 3.3.1. Triggers of Anaphylaxis in the Whole Analysed Group

We confirmed the reason of anaphylaxis in the analysed group (Figure 6)*—Hymenoptera* venom—210 cases, drugs—100 cases, food—51 cases, latex—3 cases, subcutaneous immunotherapy (SCIT)—3 cases, unknown triggers—15 cases.

#### 3.3.2. Causes of Anaphylaxis by Group Division: Women, Men, Children

The following graph and table represent the analysis of the studied groups; the factors causing anaphylaxis are shown in Figure 7 and the *p*-value between groups, taking into account the reasons, is presented in Table 2.

In many cases one trigger lead to multiple reactions from more than one system (Table 3).

#### 3.3.3. Causes of Anaphylaxis in Detail in the Whole Analysed Group

Of all 210 anaphylactic reactions following *Hymenoptera* sting, 72% were caused by wasps, 11.4% were caused by honeybees, 9.5% were caused by hornest, and 7.1% by unknown insects.

In the drug group, most cases were reported with NSAIDs (Figure 8 presents various chemical substances in the NSAIDs group), antibiotics (Figure 9 presents various antibiotics), local anaesthetic agents and single reactions after exposure to low-molecular-weight heparin, animal insulin, radio contrast media, pseudoephedrine, dextromethorphan, and tolperisone.

Foods were the cause of the reaction in 13.4% of cases. Equally frequently reactions (11%) have been reported after peanuts, tree nuts, and celery, and 5.8% after fish, honey, seafood, and egg whites. Between these two groups, we noted a 10% share of milk and 8% of various fruits. About 25% percent were single episodes after a given food (Figure 10).

#### 3.3.4. Causes of Anaphylaxis Presented in Detail in a Group of Children (n = 51)

Just as in the case of the whole group, the most common reason for childhood anaphylaxis was hymenoptera. Drugs (antibiotics and NSAIDs) and food were the causes of anaphylaxis at a similar level (Figure 11).

## 4. Analysis of Results and Discussion

The incidence of anaphylaxis was higher in women than in men (*p* < 0.000), which is consistent with other reports for Europe [11,21]. However, this is contrary to the data from Korea where anaphylaxis developed in men more often, at 65% [24]. The annual incidence per population of the West Pomerania Province falls in the range reported in other European registers (0.0015–0.0079%) [11]. However, in comparison to the data from Korea, the frequency of anaphylaxis in the group of patients reporting to Emergency Department was estimated at 0.01% [24].

The mean age at the onset of anaphylaxis was 40.5 years, which is similar to that reported in other sources—34 [25], 41.6 [26], and 44.2 years [13].

### 4.1. Clinical Manifestations

Patients presented with a range of symptoms, and there was a varied incidence of symptoms from different organ systems, but no significant differences were found between men and women (*p* > 0.05). The incidence of skin manifestations in children is statistically higher than in women (*p* = 0.0488), and children have more gastrointestinal symptoms than men (*p* = 0.0138).

Most men and women with anaphylaxis developed cutaneous (75% and 78%), cardiovascular (71% and 74%), and respiratory (59.6% and 72.5%) symptoms. However, the data acquired from the research on perioperative anaphylaxis (according to a local study in Hong Kong) indicated cardiovascular manifestation as the one which occurs most often in this group (87, 3%) [27]. Only 12% of children did not have cutaneous symptoms in the conducted research, just like in the case of Poowuttikul [28], as compared to more than 20% in adults. Carter also confirms that children with anaphylaxis had cutaneous symptoms most often [29]. Hernandez, whereas, observed respiratory symptoms in children with anaphylaxis most often [30].

It has been reported that anaphylaxis usually gives cutaneous symptoms [13,25] and cardiovascular symptoms [7]. Findings from our study are consistent with data on patients hospitalized in Karachi [25] and data from Iran [26] with respect to cutaneous symptoms of anaphylaxis, but we recorded differences in the incidence of respiratory symptoms and cardiovascular symptoms. In our study, anaphylaxis caused by *Hymenoptera* stinging was usually manifested by cardiovascular symptoms (81.4%) and cutaneous symptoms (76.7%). Patients with anaphylactic reactions triggered by drugs/food more frequently developed cutaneous symptoms than respiratory symptoms. This is consistent with data from Belgium [31], where food was reported as the most common trigger of anaphylaxis. About 40% of children had gastrointestinal symptoms compared to 21% in men’s group (*p* = 0.014).

Organ systems involved in anaphylaxis. The involvement of two and three organ systems was most commonly observed in anaphylaxis (in the whole group) in comparison to one or four organ systems (*p* = 0.000). The reaction of one and four organ systems in anaphylaxis (in the whole group) accounted for no more than 11% of cases. Clinical manifestations involving one organ system was less common in children compared to women (*p* = 0.044) and men (*p* = 0.033). Four-organ systems were involved more frequently in children than in women (*p* = 0.032).

### 4.2. Diagnostic Methods

(a) A medical interview was conducted with all patients (100% of cases); it was decisive for the inclusion of the patient in the registry. However, not all patients provided their medical documents with information supporting the reported event/reaction. 

(b) Skin prick tests are the basic test routinely performed when an acute IgE-mediated reaction is suspected, or are performed to provoke a cutaneous reaction if the sensitivity and specificity (SE, SP) of a given assay are unknown. Tests using substances for which sensitivity and specificity are unknown require at least one reference test in a group of healthy volunteers. It should be kept in mind that they need to be performed 6 weeks after the reaction takes place [32]. Skin tests were performed in 283 patients.

(c) Measurement of sIgE levels is useful when skin tests cannot be done (e.g., in patients taking antihistamine drugs), in patients presenting with dermographic urticaria, or in patients with a history of very severe anaphylactic reactions (grade IV). Considering our registry, measurements of sIgE levels were particularly useful for the determination of an insect causing anaphylaxis. In a study by Fontaine et al., the sensitivity of beta-lactam-specific IgE measurements ranged from 0% to 50%, and specificity ranged from 83% to 100% [33]. In our study, none of the patients who experienced anaphylaxis taking beta-lactams w positive for sIgE antibodies, despite positive skin tests or positive basophil activation tests (BAT 203c). The absence of sIgE in the blood of beta-lactam-allergic patients is associated with the decline in the level of these antibodies over time [34]. Levels of sIgE were measured in 238 patients.

(d) Tryptase is a marker of mast cell activation [35,36], and its levels are measured at 30, 60, and 120 min following the onset of anaphylactic reaction. It is also recommended to mark the tryptase level during anaphylaxis in children [30]. Elevated levels of tryptase (i.e., a 20% increase + 2 IU above the baseline level) are useful in differentiating types of anaphylaxis and other conditions with similar clinical symptoms [37,38,39]. A major limitation of tryptase measurement concerns food-induced anaphylaxis, because this reaction is not manifested by increased tryptase levels in blood [40,41]. Tryptase level was measured in 148 patients from the analysed group. 

(e) Challenge tests are generally not recommended in patients with a history of severe anaphylaxis, but when an anaphylactic reaction is caused by more than one potential trigger, e.g., a combination of NSAIDs and antibiotics, or the patient has perioperative anaphylaxis and additional tests were negative, then the challenge tests may be considered after the analysis of potential risks and benefits. Challenge tests were performed in 23 patients included in the registry.

### 4.3. Causes of Anaphylaxis

(a) Data for 10 years from the analysed registry also indicate that *Hymenoptera* venom was the most frequent cause of anaphylaxis. Epidemiological statistics for Italy, Germany and Austria are similar [13,42]. Different data are presented in the United Kingdom (UK), where insect stings accounted for only 32% of reasons of anaphylaxis [43]. Anaphylaxis developed after field stings and in two cases during venom immunotherapy, when the ultra-rush protocol was used. Similar cases have been reported by other authors [7,44,45,46,47] and the taskforce on *Hymenoptera* venom allergy of the European Academy of Allergy and Clinical Immunology (EAACI). In our registry, anaphylaxis most frequently was caused by wasp stings (72%), and a similar incidence was reported by NORA Members (70.4%) [13,48]. This was true for the whole analysed group and in each group separately. 

(b) Drugs are another potential cause of anaphylaxis. Findings from our analysis are similar to those for an adult population from Europe [7], the UK [43], although in 2017, Gonzalez-Estrada et al. [42] reported that drugs were only responsible for 13.3% of anaphylaxis cases. On the other hand, Khan et al. estimated the incidence of drug-induced anaphylaxis at 60% [25] and 41% [48]. In our registry, NSAIDs were the most frequent cause of nonallergic anaphylaxis (46% was drug-induced anaphylaxis), which is consistent with statistics presented by Aun et al. [49] and Jared et al. [50]. The aforementioned information does not refer to children, as in this group it was antibiotics (cephalosporins) which were the most common reason of drug-induced anaphylaxis. Most drug-induced nonallergic anaphylactic reactions recorded in our registry were induced by acetylsalicylic acid. Slightly different data were reported by researchers from Spain and Germany (NORA), where metamizole (the reason of allergic anaphylaxis) was the most frequent cause of anaphylaxis [13,21]. 

Beta-lactams are the most common antibiotics causing anaphylaxis [51,52]. The second reason of drug-induced anaphylaxis (in the whole group), in our registry, were cephalosporins, not penicillin (PN), as has usually been reported by others [13,51,53]. The analysis of our registry did not reveal significant differences between the incidence of reactions induced by penicillin and cephalosporins, probably because of the small size of the analysed groups. Anaphylaxis due to cefuroxime (cefuroxime axetil) was rarely reported [52,54], but this is not true for our registry and data from Hong Kong [27]. Other classes of antibiotics caused anaphylaxis in single patients. 

(c) In the analysed registry, we found the lowest incidence of food-induced anaphylaxis (13.4% in the whole group, 15.7% in the children’s group), which contradicts data for France, Spain, Greece, Ireland, and Switzerland, but is consistent with data for Germany, Austria and Italy [13] and Pakistan (16.3%) [25]. Some researchers reported that food was responsible for 30% [42] or even 37% of all anaphylactic reactions [55]. Some papers state that food-induced anaphylaxis is the most common reason in the children’s group [28,55,56,57,58].

In the analysed group, food-induced anaphylaxis was mostly caused by peanuts, tree nuts, and celery (11% of cases each) which is consistent with data from the German registry [11]. Similarly to the data from United States, peanuts and tree nuts are the most common food causing anaphylaxis [59]. Anaphylaxis was also triggered by milk, fruits, fish, honey, seafood, and hen egg whites, while other types of food caused reactions in single patients. In the children’s group, food-induced anaphylaxis developed after exposure to peanuts, hen egg whites, milk, tree nuts and celery, respectively. Data from our clinic regarding the incidence of milk-induced anaphylaxis are consistent with those reported by Gonzalez-Estrada et al. [42]. We also identified a high incidence of anaphylaxis due to peanuts and other nuts, although it was lower than the 17% reported by NORA Members [13] or the 20% reported in the Spanish registry [42].

(d) Latex and subcutaneous immunotherapy (SCIT) induced anaphylaxis in three patients each (0.79% of registered reactions); the incidence of latex-induced anaphylaxis in Europe is 0.3% [2,7,13]. The incidence of anaphylaxis caused by SCIT varies depending on the allergen and protocol (conventional, ultra-rush), and WAO estimated it at 0.2% of injections [60].

(e) In 15 patients (3.9%), we were unable to identify the cause of anaphylaxis, and therefore idiopathic anaphylaxis was diagnosed. It has been reported that idiopathic anaphylaxis accounts for 6.5% [13], 9.8% [24] 13.7% [42], 20% [7] and even 41% [53] of all anaphylaxis cases.

## 5. Limitations

Because this was retrospective study, it may have been influenced by selection bias. Not all patients completed all the investigations. There was a lack of detailed data on the history of anaphylaxis information from healthcare professionals about the treatment during the episode.

The population of the youngest children is under-represented in the analysed registry, since our Allergy Clinic is a reference centre for children older than 5 years, and adults.

Under-diagnosis of anaphylaxis in children may be because symptoms are not as overt, especially in infants, where the symptoms of anaphylaxis may be subtle and often noncardiovascular.

The testing of tryptase levels in our department was introduced in the middle of 2007. The group of patients with an allergy to insect venom is over-represented, due to the fact that we are the only one within a 200–300 km radius to perform venom immunotherapy.

## 6. Conclusions

Anaphylaxis is a severe and life-threatening reaction. Monitoring its range is very important to monitor changes in allergy development. Comprehensive management of patients who have had anaphylaxis should be complex, so partnership between allergy specialists, emergency medicine and primary care providers is necessary.

The main reason of each investigation, of a patient who has already experienced anaphylaxis, is to uncover the causal antigen of an anaphylactic reaction Moreover, it is crucial to stop the exposure during the reaction and/or to prevent future exposure after recovery.

## Figures and Tables

**Figure 1 ijerph-17-02787-f001:**
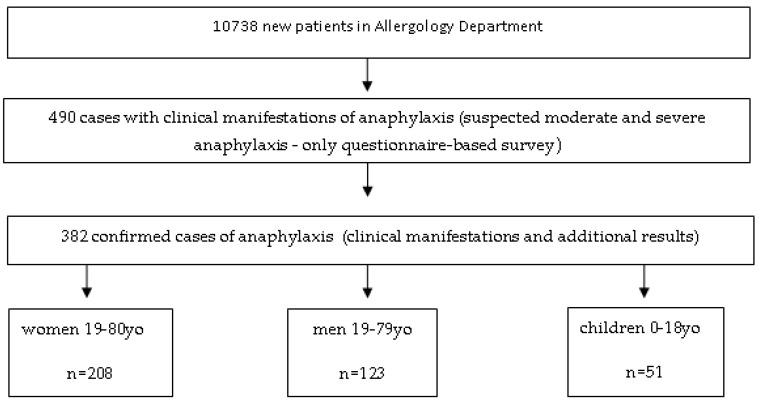
The structure of research and the division into groups.

**Figure 2 ijerph-17-02787-f002:**
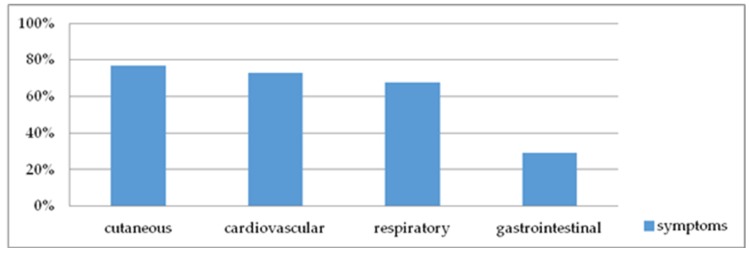
Organ systems involved in anaphylaxis.

**Figure 3 ijerph-17-02787-f003:**
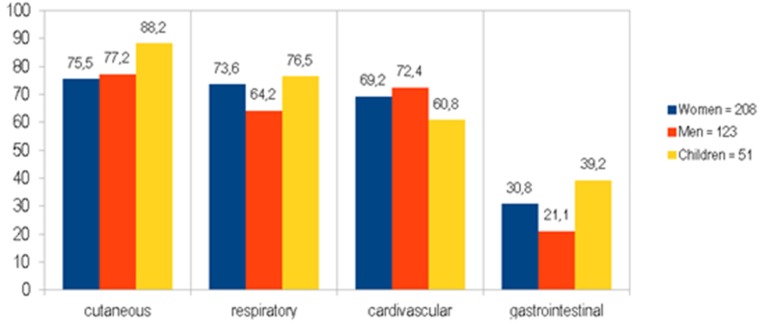
Organ systems involved (%) in anaphylaxis with groups divided into women, men, and children.

**Figure 4 ijerph-17-02787-f004:**
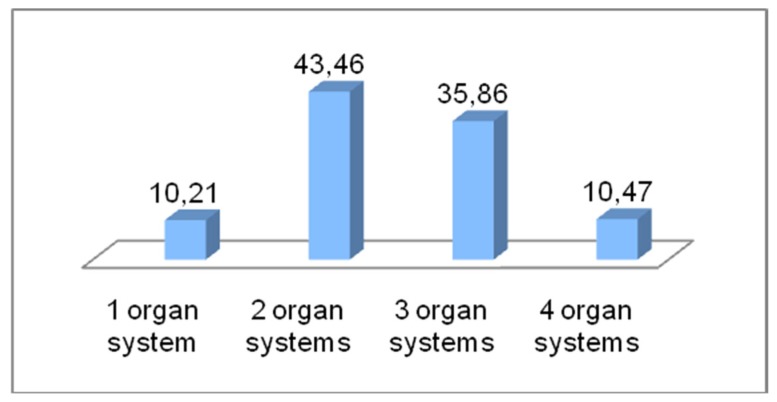
% of case anaphylaxis with 1, 2, 3 or 4 affected organ systems (in the whole group).

**Figure 5 ijerph-17-02787-f005:**
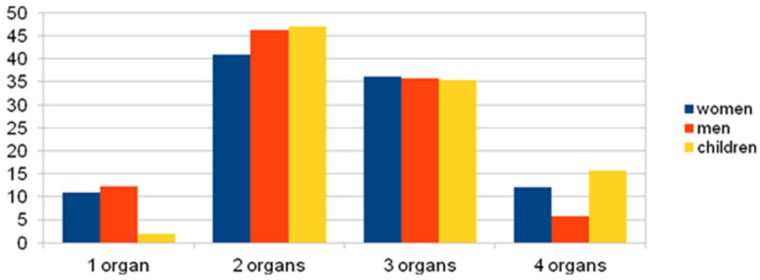
% of Cases anaphylaxis with 1, 2, 3 or 4 affected organ systems with groups divided into women, men, and children.

**Figure 6 ijerph-17-02787-f006:**
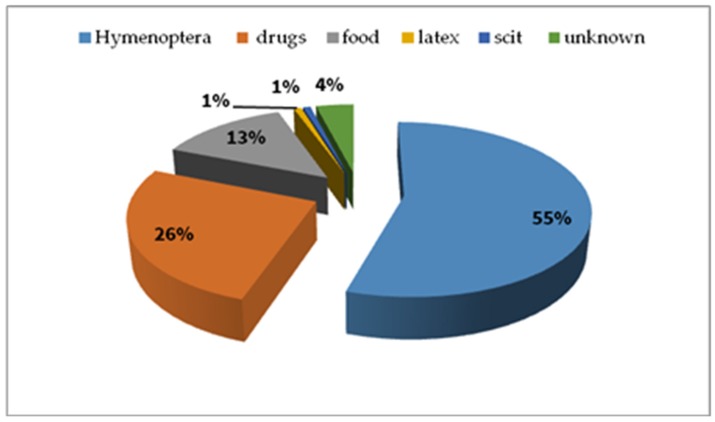
Triggers of anaphylaxis in the analysed whole group (%).

**Figure 7 ijerph-17-02787-f007:**
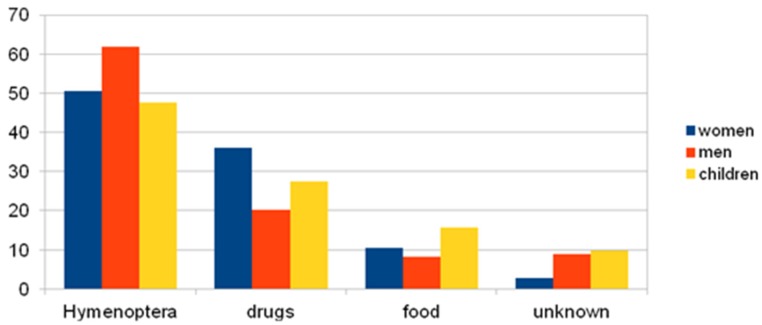
Triggers of anaphylaxis in the analysed groups of women, men, and children (%).

**Figure 8 ijerph-17-02787-f008:**
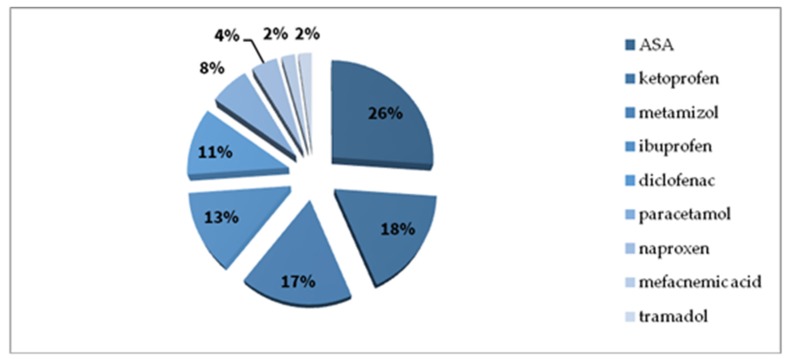
Nonsteroidal anti-inflammatory drug (NSAID) causes of anaphylaxis in the study whole group (%).

**Figure 9 ijerph-17-02787-f009:**
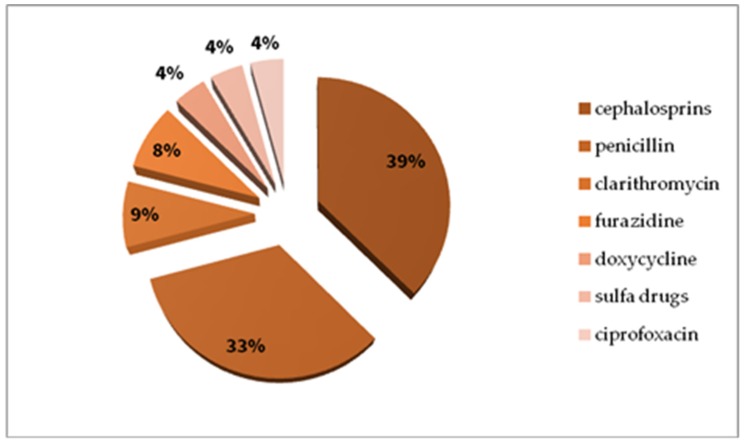
Antibiotic causes of anaphylaxis in the study whole group (%).

**Figure 10 ijerph-17-02787-f010:**
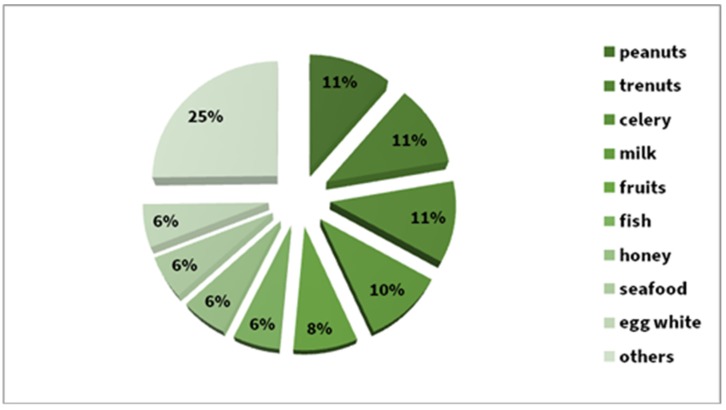
Food causes of anaphylaxis in the whole study group (%).

**Figure 11 ijerph-17-02787-f011:**
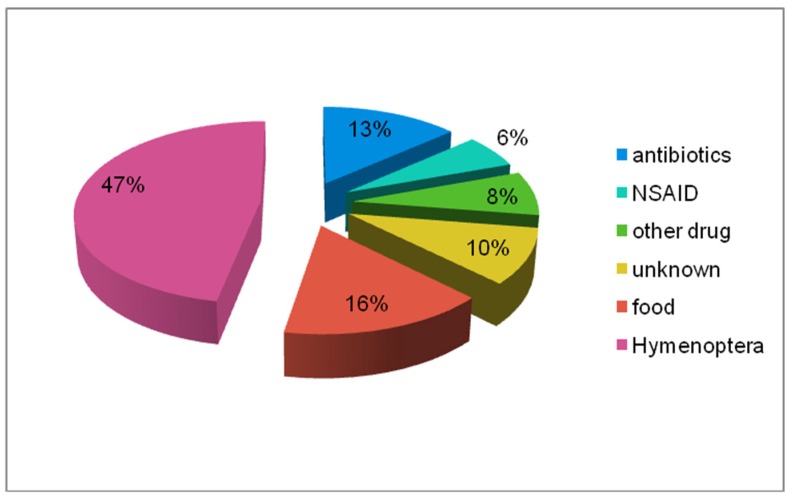
Triggers of anaphylaxis in the children’s group (%).

**Table 1 ijerph-17-02787-t001:** Classification of anaphylaxis severity [12].

**Classification by Mueller**	
**Grade I**	Generalized urticaria, itching, malaise, and anxiety
**Grade II**	Any of the above plus two or more of the following: angioedema, chest constriction, nausea, vomiting, diarrhoea, abdominal pain, dizziness
**Grade III**	Any of the above plus two or more of the following: dyspnoea, wheezing, stridor, dysarthria, hoarseness, weakness, confusion, feeling of impending disaster
**Grade IV**	Any of the above plus two or more of the following: fall in blood pressure, collapse, loss of consciousness, incontinence, cyanosis
**Classification by Ring and Messmer**	
**Grade I**	Generalized skin symptoms (e.g., flush, generalized urticaria, angioedema)
**Grade II**	Mild to moderate pulmonary, cardiovascular, and/or gastrointestinal symptoms
**Grade III**	Anaphylactic shock, loss of consciousness
**Grade IV**	Cardiac arrest, apnoea

**Table 2 ijerph-17-02787-t002:** *p*-value between groups, taking into account the reasons.

	*p*-Value			
**Triggers**	**Hymenoptera**	**Drugs**	**Food**	**Unknown**
**Women/Men**	*p* = 0.045	*p* = 0.0044	n.s.	*p* = 0.0158
**Women/Children**	Nonsignificant (n.s).	n.s.	n.s.	*p* = 0.0281
**Men/Children**	n.s	n.s.	n.s.	n.s.

**Table 3 ijerph-17-02787-t003:** Type of anaphylactic reaction vs. triggers.

% of Registered Reactions from Organs VS Trigger Factor					
Trigger	Cutaneous Symptoms	Gastrointestinal Symptoms	Respiratory Symptoms	Cardiovascular Symptoms	Other Symptoms
**Food**	90	33	61	51	4
**Drugs**	69	27	66	66	5
**Venom**	77	29	71	82	2
**Latex**	100	67	33	67	0
**SCIT**	100	33	67	67	0

Within the range of factors causing anaphylaxis most often, the following involvement has been noticed—for food, cutaneous ˃˃ respiratory ˃ cardiovascular ˃ gastrointestinal; for drugs, cutaneous ˃ respiratory = cardiovascular ˃˃ gastrointestinal; for insect venoms, cardiovascular ˃ cutaneous ˃ respiratory ˃˃ gastrointestinal. Cutaneous manifestation is the most important for anaphylaxis induced by food, drugs, latex, and allergy vaccination. In reference to insect venom, manifestations from cardiovascular system are the most important ones.

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
