# Peer review of "Clinical Manifestations and Causes of Anaphylaxis. Analysis of 382 Cases from the Anaphylaxis Registry in West Pomerania Province in Poland"

_ijerph, 2020, doi:10.3390/ijerph17082787_

Round 1

Reviewer 1 Report

MAJOR COMMENTS:
   - There is a major issue with the definition of anaphylaxis in the article:

x The time to onset of anaphylactic reactions is part of their definition, they are “immediate” (clinical signs appears within 2h of exposure). This needs to be properly explained.

x Also, anaphylactoid reactions (immediate non IgE mediated immune reactions) were included within the term “anaphylaxis”, but they should not. To keep them in the study after clearly stating that they are not anaphylactic reactions, the authors need to either use “anaphylaxis” in quotes or to use (pseudo-)anaphylaxis throughout the rest of the manuscript.

  • Not enough info in the introduction (e.g. add a section about drug allergy, add a section on contact dermatitis)

  • Not enough details in the method section:

x Subdivide into different subsections (with sub-titles) that describe the different aspects of the methods

x Provide the number of questions in each section in the text + provide the questionnaire

x Clarify when and by whom the questionnaires were filled out

x It is probable that it should have been tested with a non-parametric approach (i.e. not a T-test). This needs to be addressed.

  • Not enough data provided in the results:

x Subdivide the results into different subsections (with appropriate sub-titles)

x More could have been done with the data: e.g. female/male ratio in children; children proportion; more comparison with the general patient population of the hospital; proportion of patients with 1 vs 2 vs 3 affected organ systems; no info on onset despite the fact that it is a key element in the definition of anaphylaxis; no information on severity of symptoms/treatments/recovery; potential symptomatic or causal differences between children and adults…
x BAT is mentioned in the discussion, but not in the method or result sections.

  • Not enough references/studies used in the discussion to compare the present results

x More comparisons need to be made with the very large literature on anaphylaxis. Only a few are presented and each time with a very limited number of articles despite the large number of similar studies.
x Discussions are not usually subdivided with subtitles, unless this is required by the present journal?

  • Limitations: There are many more limitations to this study than lacking children and the technical limitations of the tryptase assay.

  • Conclusion: the main reason to uncover the causal antigen of an anaphylactic reaction is to stop its exposure during the reaction and/or to prevent future exposure after recovery.

MINOR COMMENTS
   - More references need to be added throughout the manuscript.

  • The English needs to be reviewed. Several sentences need complete rewriting.
  • The % are hard to see on the pie charts.
  • Some legends are missing from the bar graph.

All acronyms need to be explained the first time they are used in the text.

Reviewer 2 Report

Dear Authors,

I found yours article “Clinical manifestations and causes of anaphylaxis. Analysis of 382 cases from the Anaphylaxis Registry” interesting and important to develop the knowledge about allergies. You analyzed a lot of data but I have some comments and suggestions, which could improve manuscript quality.

Comments and suggestions:

Numbered list is not proper form for results describing and analysis.

Lines 68-69: It will be nice to motion that new sources of food proteins, development of new technologies in food production change the immunogenic or/and allergenic potential of final product ingredients.

Lines 93-95: This is kind of group characteristic. 382 cases were chosen for all analysis. Middle age was 38.2 for men –range from 0 to 79. In Line 228 you wrote that “our Allergy Clinic is a reference centre for children older than 5 years and adults”. Why did you take patients below 5 years? 
The same comment for women. No minimal age for children and adolescent group which actually is not a group discussed later.

Lines 96-103: finally those numbers are pictured on Figure 1 which name should be clinical symptoms of anaphylaxis without percent (according to line 96). The OX axis has no proper labels and bars have different 3D effect. OY axis has no title (probably legend should serve as a title).

Lines 107-113: is any formal or medical reason why all patients did not been diagnosed the same whey? If there is may be they should be split for minimum 6 groups according the test did? Please explain this point.

Line 114: How varies anaphylaxis scores in each trigger group? It is important to know does statistic is about grade II or IV.

Line 126: Table 2: The word number in the first row suggested that it is the number of patients who recognized those symptoms. In the second row, you put [%], please explain that. It will be nice to do statistics for these results to show the reader which symptoms are more important for each trigger group.

Line 135: Please change for ‘Analysis of results and discussion’ and remove a numbered list form.

Lines 155-184: it look for me that Material and methods will be better plase for this part.

Line 202: ‘reported by others’ will be better.

Line 212: What does it mean ‘11% of case each’? The authors present one group cold food among drugs, venom etc. How many cases it was?

Author Response

Thank You for all the comments, they were very helpful.

Lines 68-69: It will be nice to motion that new sources of food proteins, development of new technologies in food production change the immunogenic or/and allergenic potential of final product ingredients.

  1. Lines 68-69

In my version(2) - lines 85-87.

Yes , I agree that this aspect of changes taking place during food production is worth mentioning in the manuscript.

I add: New sources of food proteins, development of new technologies in food production change the immunogenic or/and allergenic potential of final product ingredients

Lines 93-95: This is kind of group characteristic. 382 cases were chosen for all analysis. Middle age was 38.2 for men –range from 0 to 79. In Line 228 you wrote that “our Allergy Clinic is a reference centre for children older than 5 years and adults”. Why did you take patients below 5 years?
The same comment for women. No minimal age for children and adolescent group which actually is not a group discussed later.

  1. Lines 93-95

In my version(2) - Line 178

The characteristics of the group, quoted by reviewer, is in the manuscript - version 1 in which the whole analyzed group is divided into men and women only. After one of the reviewers question, a division into three groups was included in the manuscript – version 2, which is shown in Fig. 1. Fig3, Fig 5, Fig 7, Fig 11.Therefore, after the first revision the age range of men and women was given in brackets. It ranges from 19 to 79 or from 19 to 80, because a group of children was separated. The youngest patient was 6 months old, so I included the age range from 0 to 18 in Fig.1. Line 170

In manuscript vers. 2 is: The mean age at the onset of anaphylaxis was 45.3 years (range 19-80) for women (n=208), and 44.3 years (range 19-79) for men (n=123). In the group of children and adolescents (max. age 18 years) the mean age at onset was 11.3 years (9.9 for boys and 12.1 for girls).

3.line 228 “ reference center for children older than 5 years”

In my version(2) line 426-427

It is a limitation in this project.

Our clinic is a reference center which provides constant medical care for older children under 5 years of age. In our medical team, there is one pediatric allergist who is not available every day for patients. There is another children’s hospital in the city where younger patients (under 5 years of age), suffering from allergies and anaphylaxis, can be admitted. I have informed doctors who work there to tell patients to consult me, but patients do not always have time or they may not be willing to do so. When a younger patient (under 5 years of age) came to our clinic and the pediatric allergist was available, the child was admitted by this doctor.  Those children are taken into account in the analysis.

 Lines 96-103: finally those numbers are pictured on Figure 1 which name should be clinical symptoms of anaphylaxis without percent (according to line 96). The OX axis has no proper labels and bars have different 3D effect. OY axis has no title (probably legend should serve as a title).

  1. lines 96-103 Figure 1
    In new version(2) it is figure 2 and line 192

It is a mistake which appeared during formatting. Axis OX shows clinical manifestation (symptoms), whereas axis OY presents the percentage of patients with a particular manifestation. I prepared figure without a 3D effect.

Lines 107-113: is any formal or medical reason why all patients did not been diagnosed the same whey? If there is maybe they should be split for minimum 6 groups according the test did? Please explain this point.

5.Lines107- 113

 In my version(2) 218- 224 the results are not discussed here, diagnostics was conducted in such a way.

Medical explanation of this fact can be found in the analysis and discussion section  - lines 340-367

Medical history was taken from all of the patients, but skin tests, as an example of in vivo diagnostic tests, are risky as they might cause anaphylaxis. In the situation when the patient is afraid to have such tests done, they are not done at all – it is described in the analysis of the results. Then, sIgE was the first choice method as a safe examination - in vitro. Sometimes, however, despite positive skin tests, it was impossible to indicate which insect was the trigger of anaphylaxis: a wasp or a bee (double positivity). Then, in addition to that, sIgE is indicated by means of recombinant allergens. As I have mentioned in the limitations of the article, the level of tryptase could be indicated at the end of 2007. That was only possible after the acceptance of hospital authorities, in justified cases. Tryptase tests in Poland are expensive, national insurance does not cover the costs. Provocation tests, described in the discussion, are contraindicated in a patient after severe anaphylaxis. They are for selected cases. To be more specific, they are dedicated for perioperative anaphylaxis or for small children after getting a negative result of scratch test and sIgE.

Thus, too many examinations should not be done if we get a confirmation at the first stage of additional tests – I do not burden the budget or the patient with costs. There is no need to divide patients into six groups according to the diagnostic methods used.

Line 114: How varies anaphylaxis scores in each trigger group? It is important to know does statistic is about grade II or IV.

6.line 114

Unfortunately, the application form in the information system designed by a computer scientist was flawed and it grouped particular symptoms within one system. Patients’ data is anonymous, I cannot reach the source material now. I am not able to state the level of anaphylaxis severity.

I implemented statistics describing the involvement of one or more organ systems during anaphylaxis.

Lines 328-334

 Line 126: Table 2: The word number in the first row suggested that it is the number of patients who recognized those symptoms. In the second row, you put [%], please explain that. It will be nice to do statistics for these results to Line 126 table 2

In new version(2) it’s table 3, Line 243
Yes, I agree it’s my mistake during translation. Number refers to a quantity expressed in % .

That is why I suggest: % of registered reactions from organs vs trigger factor

I suggest to add Lines 258-265

Within the range of factors causing anaphylaxis most often, the following involvement has been noticed:

For food: cutaneus ˃˃ respiratory ˃ cardiovascular ˃ gastrointestinal

For drugs: cutaneus ˃ respiratory = cardiovascular ˃˃ gastrointestinal 

For insect venoms: cardiovascular ˃ cutaneous ˃ respiratory ˃˃ gastrointestinal

Cutaneous manifestation is the most important for anaphylaxis induced by food, drugs, latex, allergy vaccination. In reference to insect venom, whereas, manifestation from cardiovascular system is the most important one.    

Line 135: Please change for ‘Analysis of results and discussion’ and remove a numbered list form.

Line 135

In new version (2) Line 298

Yes, I will delete the numbers from the discussion, they are not necessary, technical remark

I introduced them at the beginning to put everything in order.

Lines 155-184: it look for me that Material and methods will be better plase for this part.

Line 155-184

I think that placing Material and methods (technical details) in discussion will not be a good change.

I cannot agree with that.

Line 202: ‘reported by others’ will be better.

Line  202

In new version (2) – Line 394

It is a change of a word, I agree with that „ it has usually been reported by others”

Line 212: What does it mean ‘11% of case each’? The authors present one group cold food among drugs, venom etc. How many cases it was?

Line 212

In new version (2) Line 405- 406

It is a translation mistake; only food-induced anaphylaxis was analyzed here (the whole group is 51 people and that constitutes 100%)

In the analysed group (n=51), food-induced anaphylaxis was mostly caused by peanuts, tree nuts, and celery (11% of cases each)…

Reviewer 3 Report

This review is rather old-fashioned, and does not include novel findings or knowledge. The frequency of hypnoptera appeared too high. Is there any bias for the data?

Author Response

Thank You for the comments.

Reviewer 4 Report

Line 7 and 8 in Abstract must be modified, sentence too vague

The Combining of Results and discussion makes readability difficult.

make the Results comprehensive.

For Discussion

Set up as headings for discussion and re-state Findings and what is confirms in the literature and if any conflicits.

Cannot be accepted till Discussion and Results are completely redone

Author Response

Thank you for all the comments. They were very helpful.

Round 2

Reviewer 3 Report

The topic itself and items for analyses are too ordinary, and the manuscript does not show novel findings or knowledge. The authors should analyse different aspects on anaphylaxis, such as biomarkers.  I recommend to submit this manuscript to the journal with lower IF.

Author Response

Thank You for Your comments. I wish You a Happy Easter

The presented research on anaphylaxis concerns the population from northwestern Poland. So far, there has been no data referring to this significant country from Central Europe. Moreover, no central, national anaphylaxis register is kept. The present project is carried out by the Pomeranian Medical University and it is used for the  collection of data, observation of the evolution of anaphylaxis and allergies in the future. In this situation it cannot be based on markers, because I do not have biological material; I did not take part in anaphylaxis.

It is a clinical, retrospective work in which I can analyze cases of past anaphylaxis. In addition to this, on the ground of recommendations from Uppsala University, I am able to assess moderate and high probability that the description given by a patient or a doctor, who is not an allergist, refers to anaphylaxis, and not any other disease. 

To sum up, markers of anaphylaxis can be examined and assessed during anaphylaxis (if we witness the anaphylactic reaction in Emergency Department).

The research is of cognitive value. There are various causes of anaphylaxis in different geographical areas. We should bear in mind all the issues associated with anaphylaxis so that there would be no problems with diagnosing it. Such a situation, the case of severe anaphylaxis, which shows no cutaneous manifestations is very difficult to recognize. It still happens that blood samples are not taken in order to mark tryptase and patients do not get adrenaline despite the person was diagnosed with symptoms of anaphylaxis. Patients are in neurological and cardiological wards, they undergo toxicology testing, because an emergency worker / a doctor does not notice a wasp or bee sting. In hospitals during the preparation before a surgical procedure, iatrogenic mistakes are made as health service workers give medicine from the group that a patient reacted to. How many unexplained deaths due to anaphylactic reactions are there? We do not know that, because anaphylaxis is not stated as the cause of death. Doctors state cardiorespiratory failure.Thus, it is worth spreading the knowledge about anaphylaxis. In literature, there are registers from different countries which analyze, among others, the causes of anaphylaxis and clinical manifestations.

Reviewer 4 Report

Much improved; 

The Title should reflect your country, or regional area.

Author Response

Thank You for Tour support. I wish You Happy Easter.

Yes, I agree it’s my mistake. I change this:
Clinical manifestations and causes of anaphylaxis. Analysis of 382

cases from the Anaphylaxis Registry in West Pomerania Province in Poland